# Image to Icosahedral Projection for SO(3) Object Reasoning from Single-View Images

**David Klee**                                              KLEE.D@NORTHEASTERN.EDU
**Ondrej Biza**                                             BIZA.O@NORTHEASTERN.EDU
**Robert Platt**                                            R.PLATT@NORTHEASTERN.EDU
**Robin Walters**                                       R.WALTERS@NORTHEASTERN.EDU
*Northeastern University, Boston, MA, USA*

**Editors:** Sophia Sanborn, Christian Shewmake, Simone Azeglio, Arianna Di Bernardo, Nina Miolane

## Abstract

Reasoning about 3D objects based on 2D images is challenging due to variations in appearance caused by viewing the object from different orientations. Tasks such as object classification are invariant to 3D rotations and other such as pose estimation are equivariant. However, imposing equivariance as a model constraint is typically not possible with 2D image input because we do not have an a priori model of how the image changes under out-of-plane object rotations. The only SO(3)-equivariant models that currently exist require point cloud or voxel input rather than 2D images. In this paper, we propose a novel architecture based on icosahedral group convolutions that reasons in SO(3) by learning a projection of the input image onto an icosahedron. The resulting model is approximately equivariant to rotation in SO(3). We apply this model to object pose estimation and shape classification tasks and find that it outperforms reasonable baselines.

**Keywords:** equivariance, pose estimation, shape classification

## 1. Introduction

In many applications such as robotic manipulation, autonomous vehicle navigation, scene segmentation, or 3D modeling, it is useful to reason about the 3D geometry of objects in the world using sensor data captured from only a single point-of-view. Such applications have thus motivated much work in computer vision dedicated to inferring 3D information such as object orientation, position, and size (Mildenhall et al., 2020; Wang et al., 2019; Li et al., 2019; He et al., 2020; Xiang et al., 2017) from images. These methods, however, fail to capture and exploit the symmetry of 3D space. While some computer vision models incorporate 2D symmetry (Weiler and Cesa, 2019), it is very difficult to incorporate 3D symmetries.

Alternatively, SO(3)-equivariant neural networks have emerged as a powerful tool for modeling 3D geometry. By leveraging the 3D symmetries of the data, these networks demonstrate improved generalization properties, data efficiency, and provable consistency (Elesedy and Zaidi, 2021; Batzner et al., 2022; Fuchs et al., 2020). To apply such symmetry-aware methods to pose estimation, however, the input data must be SO(3)-transformable and thus, until now, they have been limited to either point clouds (Chen et al., 2021), spherical images (Esteves et al., 2018), or highly structured multi-view images (Esteves et al., 2019).

We propose Image2Icosahedral (I2I), a method that allows for SO(3)-equivariant reasoning on embeddings learned only from single-view image inputs. The embeddings and downstream model can be trained in an end-to-end manner. Since our model incorporates 3D symmetries, it is able to generalize better than other image-based models over transformations induced by changes in camera viewpoint or object orientation.

I2I has two novel components which separate the problems of feature extraction and object-level memory, (1) a learnable projector which maps images to spherical dynamic filters and (2) a dynamic filter icosohedral convolution. The spherical filter projector maps 2D images onto features defined over 6 vertices of an icosohedron, representing the partially observed object features and the camera viewpoint. We implement this projector using an E(2)-equivariant CNN Weiler and Cesa (2019). Thus the projector does not need to directly reason about how the object looks from other angles. The output of the projection is then used as a dynamic filter which is convolved over the set of vertices of the icosohedron with learned object feature maps. The object feature maps are signals over the entire icosohedron and thus represent, in latent space, the fully observed object in an explicitly SO(3)-transformable manner.

To summarize, our contributions are: (1) we propose a novel model that implements a discrete approximation to an SO(3)-equivariant dynamic convolution filter over $S^2$; (2) we propose a novel method of projecting a 2D image onto an equivariant spherical convolutional filter; (3) we demonstrate I2I outperforms relevant baselines on object orientation prediction, by an order of magnitude for some objects, and shape classification tasks.

## 2. Related Work

**3D Shape Analysis** Research into 3D object recognition has been facilitated by large datasets of common object models (Wu et al., 2015; Chang et al., 2015; Uy et al., 2019; Fu et al., 2021). The recognition problem has been studied for several input modalities including multiview images (Su et al., 2015), point clouds Qi et al. (2017), and projections of 3D geometry Bai et al. (2016); Cohen et al. (2018). In the challenging setting where objects are not aligned to a canonical orientation, methods that reason about 3D space can achieve better performance. Kanezaki et al. (2018) implicitly reason about the image view point, while EMVN (Esteves et al., 2019) uses structured view points so that information can be processed in an SO(3) equivariant manner. In contrast, our method reasons about 3D rotation using a single image input with no restrictions on the camera view point.

**6D Pose Estimation** Reasoning about the 3D position and orientation of objects in a single image is challenging, yet has important applications in robotics (Tremblay et al., 2018). Pose estimation has also been approached using convolutional networks which are trained to predict the 3D bounding boxes (Xiang et al., 2017), visual keypoints (He et al., 2020; Manuelli et al., 2019) or 3D object coordinates (Wang et al., 2019; Li et al., 2019; Zakharov et al., 2019), with which the pose can be extracted based on known properties of the objects such as 3D size, appearance, or canonical instance. In contrast, our method does not require any information about the object, allowing it to generalize to novel instances at test time, and it explicitly incorporates 3D rotational symmetry.

**Rotation Equivariance** Neural networks with equivariance to 3D rotations (the SO(3) abstract group) have been effectively applied to the tasks of shape classification and pose estimation for 3D objects. Many unique approaches exist for processing spatial data (e.g. point clouds) using the continuous rotation group (Thomas et al., 2018; Deng et al., 2021; Fuchs et al., 2020) and the discrete Icosahedral group (Chen et al., 2021). Other works applied spherical convolution and icosahedral convolution to 3D images (Cohen et al., 2018) and multi-view images (Esteves et al., 2019), respectively. These approaches differ from our work, because the input can be acted upon by a 3D transformation, whereas we use 2D images where such a transformation is not available. Similar to our work, Quessard et al. (2020) and Park et al. (2022) proposed networks that extract SO(3) equivariant latent states from images. However, they were applied to simple tasks, such as pose prediction of a single object.

## 3. Background

### 3.1. Equivariance

The concept of equivariance to group transformations formalizes when a map preserves symmetry. A symmetry group is a set of transformations that preserves some structure. Given symmetry group $G$ that acts on spaces $\mathcal{X}$ and $\mathcal{Y}$ via $\mathcal{T}_g$ and $\mathcal{T}'_g$, respectively, for all $g \in G$, a mapping, $f \colon \mathcal{X} \to \mathcal{Y}$ is equivariant to $G$ if

$$f(\mathcal{T}_g x) = \mathcal{T}'_g f(x). \tag{1}$$

That is, an equivariant mapping commutes with the group transformation. Invariance is a special case of equivariance when $\mathcal{T}'_g$ is the identity mapping.

### 3.2. Group convolution over Homogeneous Spaces

The group convolution operation is a linear equivariant mapping which can be equipped with trainable parameters and used to build equivariant neural networks (Cohen and Welling, 2016). Convolution[1] is performed by computing the dot product of a signal with a filter that is shifted across a range of shifts. In standard 2D convolution, the shifting corresponds to a translation in pixel space. Group convolution (Cohen and Welling, 2016) generalizes this idea to arbitrary symmetry groups, with the filter transforming by elements of the group. Let $G$ be a group and $\mathcal{X}$ be a homogeneous space, i.e. a space on which $G$ acts transitively, for example, $\mathcal{X} = G$ or $\mathcal{X} = G/H$ for a subgroup $H$. We can compute the group convolution between two functions, $f, \psi \colon \mathcal{X} \to \mathbb{R}^k$, as follows:

$$[f \star \psi](g) = \sum_{x \in \mathcal{X}} f(x) \cdot \psi(\mathcal{T}_g^{-1} x). \tag{2}$$

Note that the output of the convolution operation is defined for each group element $g \in G$, while the inputs, $f$ and $\psi$, are defined over a homogeneous space of $G$. By parameterizing either $f$ or $\psi$, group convolution may be used as a trainable layer in an equivariant model.

---

1. Technically, this is group correlation, but is often called convolution in the context of neural networks.

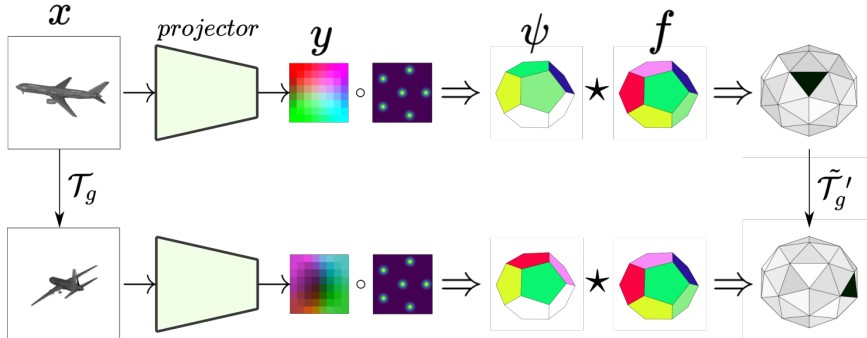

Figure 1: Illustration of our proposed model, Image2Icosahedral (I2I). We project from the input image $x$ onto a feature map $y$ using a $C_4$-equivariant encoder. Then, we project $y$ onto $\psi$, a dynamic filter onto a discrete approximation of the 2-sphere. Finally, we convolve $\psi$ with a feature map $f$ using icosahedral group convolution. The output is approximately equivariant over the icosahedral group $I_{60} \subset \mathrm{SO}(3)$.

### 3.3. Icosahedral Group

For reasoning about physical objects, it is desirable to learn functions that are equivariant to SO(3) (the group of 3D rotations). However, to make the group convolution operation computationally tractable, discrete subgroups can be used to achieve approximate equivariance to continuous groups. Moreover, discrete subgroups are compatible with common pointwise activations, and there is evidence they may be easier to optimize over (Weiler and Cesa, 2019). Unlike SO(2), which contains arbitrarily fine subgroups $C_n$, for SO(3), the largest discrete subgroup which is not contained a planar rotation group is the icosahedral group $I_{60}$ which is composed of 60 group elements. The group can be conceptualized as the orientation-preserving symmetries of a regular icosahedron (12 vertices, 20 faces) or its dual polyhedron, the regular dodecahedron.

In our work, we perform group convolution between signals that live on a quotient space of the icosahedral group, i.e. an $I_{60}$-homogeneous space. Specifically, we use the quotient space $V_{12} = I_{60}/C_5$, which arises from the action of $I_{60}$ on the twelve vertices of the icosahedron. The quotient space $V_{12}$ is a discrete approximation of the 2-sphere.

### 4. Method

The Image2Icosahedral method (I2I) is designed to solve tasks which require reasoning about 3D geometry using only 2D images as input. Our method has two parts: (1) a trainable projector from the image $x$ onto a dynamic convolutional filter over the sphere $S^2$; and (2) convolution of this filter over $S^2$ with a trainable feature defined over the sphere. The projector maps the 2D image into a 3D representation which we know how to transform using SO(3). This enables us to use SO(3)-equivariant convolution over $S^2$ to then solve the task. The projector uses an SO(2)-equivariant ResNet to project onto $S^2$ (the pipeline going from $x$ to $y$ to $\psi$ on the left side of Figure 1). For the convolution, we apply equivariant group convolution over $S^2$ of the dynamic filter with a class-specific object representation over the sphere (the convolution between $\psi$ and $f$ on the right side of Figure 1).

### 4.1. SO(2)-Equivariant Projector onto a 2D Feature Map

The projector part of our model maps an image $x$ onto a dynamic filter $\psi$ defined over $S^2$. Since SO(3) can transform signals on $S^2$, this enables SO(3)-equivariance downstream. The projector has two parts. First, a fully convolutional model maps the input image $x$ to a 2D feature map $y$ using an SO(2) equivariant model. Then, we project that map onto $S^2$. We encode the input image $x$ to a 2D feature map $y$ using a ResNet-style architecture (He et al., 2016). Since the input images do not have 3D rotational symmetry, we cannot impose SO(3)-equivariance on this part of the model. However, we can enforce symmetry with respect to SO(2) rotations in the image plane. That is, rotating the object along the roll axis of the camera corresponds to SO(2) rotations of the pixels in the image plane. We impose SO(2) equivariance using steerable convolutional layers (Weiler and Cesa, 2019).

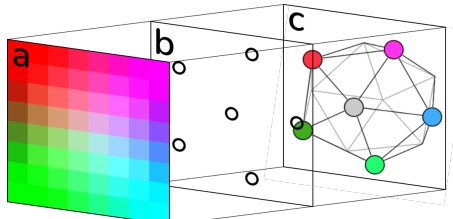

Figure 2: Projection of features from image space to quotient group of $I_{60}$. (a) Dense feature map visualized as 3-channel image; (b) Icosahedron vertices orthographically projected onto image space; (c) Resulting features on homogenous space $V_{12}$. Note that only non-occluded vertices are assigned features during projection.

### 4.2. Projection onto the Vertices of the Icosahedron

The learned 2D feature map $y$ is then orthographically projected onto $S^2$ to give a feature map $\psi$ on $S^2$ which we will use as a dynamic filter. In practice, it is necessary to discretize $S^2$ to encode features. Recall that $V_{12} = I_{60}/C_5$ denotes the quotient space which is a discrete approximation of $S^2$ over the 12 vertices of the icosahedron. We use this set as a discretization of $S^2$ since it forms a regular mesh and is compatible with the action of the icosahedral group $I_{60}$. Therefore, the dynamic filter is the map $\psi\colon V_{12} \to \mathbb{R}^k$. In practice, we achieve higher spatial resolution in the filter by parametrizing it over a submesh of $V_{12}$ (Cohen et al., 2019), which contains 42 vertices (additional vertices are placed at the midpoint of the 30 edges of an icosahedron). See Appendix B.1 for an ablation study on the benefits of using the submesh.

Similar to 2D convolutional filters, our dynamic filter is localized in that it is 0 except on the side of the sphere facing the camera. The non-occluded vertices are projected onto the image plane of the feature map, such that the identity group element is located in the center. Features for the vertices are generated by applying a Gaussian kernel centered at the vertices' projected pixel location to the feature map. The projection is illustrated in Figure 2. We find this projection scheme is effective since it preserves spatial information about the location of features in the image.

### 4.3. Dynamic Filter Icosohedral Convolution

The dynamic filters $\psi$ generated by the projector may now be used to solve the given task using SO(3)-equivariant group convolution. In group convolution, trainable parameters can be used in either the signal or filter (Jia et al., 2016). Since the learned features are locally supported on $V_{12}$, we treat it as a dynamic filter and convolve it over a signal parametrized by trainable weights. We refer to this learnable signal over $V_{12}$ as the feature sphere, since it contains features over the discretized 2-sphere.

After projection, the features are converted into a dynamic filter. That is, the vectors at each vertex are reshaped into matrices so that the learned feature maps $\psi \colon V_{12} \to \mathbb{R}^k$ become $\psi \colon V_{12} \to \mathbb{R}^{m \times n}$ assuming $k = nm$. The convolution operation between the filter and the feature sphere $f : V_{12} \to \mathbb{R}^n$ is defined:

$$[f \star \psi](g) = \sum_{v \in \mathcal{N}} f(\mathcal{T}_g v) \cdot \psi(v) \tag{3}$$

where $\mathcal{N}$ is a subset of $V_{12}$ that defines the local neighborhood around the identity element where the filter is non-zero (i.e. all non-occluded vertices during projection). This formulation can be seen to be equivalent to Eqn. 2 by re-indexing the sum $w = \mathcal{T}_g v$, but it is computationally faster since it avoids multiplying by elements $v$ outside the support of the filter. The result of this operation is a vector-valued function over the full icosahedral group, that is $[f \star \psi] : I_{60} \to \mathbb{R}^m$. For the orientation prediction task, normalizing the output using softmax gives a probability distribution over orientations of the input object.

For certain applications (e.g. object classification), it is desirable to learn features that are invariant to the rotation of the object or the image view point. An invariant representation can be achieved with a group pooling opertion, e.g. taking max (or average) along the dimension of the group.

### 4.4. Reasoning Over Continuous Orientation

In many applications, for example object pose estimation, we need to predict arbitrary rotations from SO(3). However, because our method uses discrete group convolution, it can only output signals that live on elements of the Icosahedral group. We can make predictions over the continuous group by adopting a classification-then-regression approach. First, we predict which of the discrete group elements are closest, then generate a continuous rotational offset from the selected group element. We adopt the loss function used by Chen et al. (2021) to optimize our model for continuous rotation predictions:

$$\mathcal{L}(g, R_{\hat{g}}^{\Delta}, p(\hat{g}), \hat{R}_{\hat{g}}^{\Delta}) = \mathcal{L}_{cls}(\delta_g(\hat{g}), p(\hat{g})) + \lambda \mathcal{L}_2(R_g^{\Delta}, \hat{R}_g^{\Delta}) \tag{4}$$

where $g$ is the closest element in $I_{60}$ to the ground truth orientation with ground truth offset $R_g^{\Delta}$. The first term in Equation 4 is the cross entropy loss between the predicted distribution $p$ and the delta distribution at $g$. The second term is an L2 loss on the elementwise difference between the two offset rotation matrices. In practice, we produce the rotational offset at each group element using the Gram-Schmidt orthonormalization process proposed by Falorsi et al. (2018).

## 5. Experiments

We use a ResNet-18 convolutional architecture (He et al., 2016) for the spherical filter projector, with $C_4$-symmetric kernels using e2cnn (Weiler and Cesa, 2019). The ResNet outputs a dense, 8-by-8 feature map, which is projected onto a submesh of the icosahedron (42 vertices in total; 12 original vertices plus subdivisions on the 30 edges). For the pose estimation task, our model is fully convolutional and outputs an $\mathbb{R}^7$ feature vector for each element of the icosahedral group (see Section 4.4). For the shape classification task, we perform a group pooling operation and output a vector of class labels.

We train our model using SGD with Nestorov momentum of 0.9 with a batch size of 64. The initial learning rate is $1e-3$ and decays with steps of 0.1 to $1e-5$ during training. Our method is trained for 40 epochs for orientation prediction and 20 epochs for shape classification. We augment images during training with random translations of $\pm 3$ pixels.

We evaluate our method on objects from ModelNet40 (Wu et al., 2015) and ShapeNet55 (Chang et al., 2015), using separate object instances in the training and test sets. We render 60 images of each object instance from random SO(3) views. See Appendix A.1 for additional details on dataset preparation.

### 5.1. Inferring Object Orientation

**Experiment:** Inferring object orientation in SO(3) for novel in-class objects from a single image is a challenging problem for which a variety of methods exist. We compare our method to several baselines on single view renderings of objects from ModelNet40 and ShapeNet55 (this requires generalizing predictions to novel instances within a class). We compare against two approaches for effectively predicting 3D rotations with neural networks: the Gram-Schmidt process (CNN+GS) (Zhou et al., 2019) and the Procrustes method (CNN+Proc.) (Brégier, 2021). Other competitive baseline methods combine classification and regression to predict the sign and magnitude of quaternions (CNN+$S^3_{exp}$) (Liao et al., 2019), or use a residual-like network head to refine the rotation iteratively (CNN+IER) (Bukschat and Vetter, 2020). Lastly, we include a comparison to an network that is equivariant to in-plane rotations that was suggested by Falorsi et al. (2018) (E2CNN-Eq). For some ModelNet40 objects, we include results for two methods that use partial point clouds: KPConv (Thomas et al., 2019) and EPN (Chen et al., 2021). See Appendix A.2 for more details on the implementations.

**Results and Discussion:** Tables 1 and 2 show results on ModelNet and ShapeNet objects, respectively. We find that I2I outperforms all image-based baselines on all ModelNet object classes. We notice the largest outperformance on objects with some front/back ambiguity, such as cars and benches. Because I2I performs a classification then regression, it is better suited for ambiguity, whereas methods that only perform regression will incur large penalties if they are off by 180 degrees. Surprisingly, I2I performs comparably to EPN, an end-to-end equivariant method, indicating that it is able to leverage SO(3) equivariance despite the 2D image input. In fact, on the car class, I2I achieves an error of 5.4 degrees while the point cloud methods sit at over 90 degrees; we hypothesize that the CNN encoder is more effective at incorporating global information to resolve the front/back ambiguity than the point convolution methods used on point cloud data.

| input | method | desk | bottle | sofa | toilet | car | chair | stool | airplane | guitar | bench |
|---|---|---|---|---|---|---|---|---|---|---|---|
| Grayscale | CNN+GS | 92.4 | 5.8 | 44.7 | 27.3 | 50.4 | 22.1 | 19.4 | 10.6 | 76.1 | 84.4 |
| | CNN+Proc. | 89.4 | 5.0 | 34.8 | 19.8 | 40.8 | 19.5 | 17.3 | 8.2 | 52.6 | 89.3 |
| | CNN+$S^3_{exp}$ | 103.9 | 7.8 | 31.6 | 27.5 | 57.4 | 23.7 | 24.6 | 12.3 | 40.4 | 123.5 |
| | CNN+IER | 92.9 | 6.8 | 47.3 | 30.1 | 61.5 | 20.4 | 17.8 | 10.6 | 42.3 | 87.2 |
| | E2CNN-Eq | 92.7 | 4.1 | 19.7 | 15.4 | 99.5 | 17.1 | 15.7 | 5.7 | 29.6 | 115.5 |
| | I2I (ours) | **63.1** | **2.3** | **5.2** | **6.6** | **5.4** | **7.9** | **7.3** | **2.9** | **6.5** | **18.5** |
| Point Cloud | KPConv | - | **2.0** | 16.9 | - | 113.4 | 14.6 | - | **1.54** | - | - |
| | EPN | - | 15.4 | **3.01** | - | **93.2** | **3.2** | - | 4.4 | - | - |

Table 1: Median rotation error (deg) for single-class object orientation prediction. Results are separated based on the input modality: grayscale images and partial point clouds. Bolding indicates lowest error in a given input modality.

While our method still outperforms the baselines on ShapeNet objects overall, the gap is smaller than on ModelNet. Because the ShapeNet objects are textured, this setting may be a more difficult learning problem.

| method | guitar | bed | bottle | bowl | clock | chair | file-cabinet | airplane |
|---|---|---|---|---|---|---|---|---|
| CNN+GS | 12.6 | 70.4 | 11.6 | 12.9 | **22.1** | 41.3 | 42.4 | 17.8 |
| CNN+Proc. | 17.3 | 73.8 | 15.8 | 27.6 | 39.0 | 21.9 | 39.6 | 13.8 |
| CNN+$S^3_{exp}$ | 18.3 | 72.3 | 18.2 | 32.8 | 41.5 | 22.6 | 41.3 | 14.0 |
| CNN+IER | 12.6 | 83.7 | 10.4 | **12.0** | 26.8 | 39.4 | 41.4 | 17.8 |
| E2CNN-Eq | 14.6 | 58.1 | 14.2 | 25.6 | 34.3 | 17.9 | 34.8 | 10.0 |
| I2I (ours) | **9.5** | **57.3** | **10.1** | 21.9 | 30.4 | **11.2** | **25.7** | **6.5** |

Table 2: Median error (°) on object orientation prediction task. Object classes were selected from ShapeNet55 dataset and rendered as RGB images from random SO(3) views.

## 5.2. Shape Classification

**Experiment:** In shape classification, the task is to infer the category of an object given a point cloud or image of its appearance or shape. This task is challenging because the object is presented in an orientation selected uniformly at random. We compare our method with standard baselines in four categories (Table 3). In the first two categories, we classify shape based on a single depth or grayscale image, respectively. Here, we baseline against a standard ResNet backbone (CNN) and a $C_4$-invariant ResNet model (E2CNN-Inv). We also compare against some multi-view methods, RotationNet-20 (Kanezaki et al., 2018) and EMVN-12 (Esteves et al., 2019). These methods are not directly comparable to ours because I2I takes only a single image as input. Finally, we baseline against PointNet++ (Qi et al., 2017), KPConv (Thomas et al., 2019), and EPN (Chen et al., 2021). These baselines have an unfair advantage because they take *complete, i.e. unoccluded* point clouds as input.

**Results:** Table 3 shows our results. The key observation to make is that our method (I2I) outperforms the two baslines (CNN and E2CNN-Inv) for classification from single depth images and single grayscale images (the first two categories in Table 3). The fact that I2I outperforms E2CNN-Inv is particularly significant because it suggests that our method can reason about rotation symmetry beyond the planar rotation symmetry present in the image. Surprisingly, we find that our method, when trained on depth images, can even outperform RotationNet-20 (which has access to multiple views of the same object) and PointNet++

(which has access to a complete point cloud) in terms of mean average precision. These results suggest that our model is able to leverage its ability to reason in SO(3) to improve its classification of objects presented in novel orientations.

| Input | Method | Acc. | mAP |
|---|---|---|---|
| Single Depth | CNN | 76.5 | 65.5 |
| | E2CNN-Inv | 80.4 | 70.7 |
| | I2I (ours) | 81.5 | 74.5 |
| Single Gray | CNN | 70.0 | 57.8 |
| | E2CNN-Inv | 75.1 | 64.3 |
| | I2I (ours) | 76.4 | 67.8 |
| Multi-View | RotationNet-20 | 80.0 | 74.2 |
| | EMVN-12 | 88.5 | 79.6 |
| Point Cloud | PointNet++ | 85.0 | 70.3 |
| | KPConv | 86.7 | 77.5 |
| | EPN | 88.3 | 79.7 |

Table 3: ModelNet40 shape classification results. The methods are separated by input modality: single depth image, single grayscale image, multi-view images, and a full point cloud. We report percent accuracy (Acc.) and mean average precision (mAP).

## 5.3. Ablation Study

We perform an ablation of I2I in Table 4 by separately removing the SO(2)-equivariance of the encoder and the SO(3) equivariance of the icosahedral group convolution. We find that both components provide some benefit, more so when trained on fewer views. Error increases only slightly without SO(2)-equivariance in the encoder, likely because the icosahedral convolution already preserves some SO(2) symmetry. In Appendix B.1, we compare against additional variations of our method, demonstrating that our proposed projection scheme is most effective.

| | Average Median Error (°) | |
|---|---|---|
| | 60 views | 15 views |
| I2I | 13.5 | 16.7 |
| w/o E2CNN | 15.2 | 17.5 |
| w/o GroupConv | 18.3 | 45.5 |

Table 4: Median orientation accuracy for ablations of I2I, averaged over the 10 object classes from ModelNet40. Equivariant layers provide data efficiency, creating more of a benefit when trained on fewer views.

## 6. Conclusion

We present a novel method, Image2Icosahedral, for learning 3D representations of objects from single-view 2D images. This 3D representation allows us to apply SO(3)-equivariant techniques to tasks such as orientation inference and shape classification even for image inputs. Our model is limited by the task to relying on learned symmetry in the icosahedral filter projector as the group action is unknown in image space. Since we give the ground truth orientation of objects as a point estimate and our model can only output a distribution

over $I_{60}$ and not SO(3), we cannot perfectly describe ambiguities in orientation for objects with symmetry such as a bottle. In future work, we plan to address this by explicitly returning a distribution over SO(3), effectively giving our model the ability to estimate orientation and implicitly discover symmetry in objects.

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

## Appendix A. Implementation Details

### A.1. Dataset Preparation

For ModelNet40 objects, we follow the original the train-test split of 9,843 training objects and 2,468 test objects (Wu et al., 2015), using the object models manually aligned by Sedaghat et al. (2016). The objects are rendered as grayscale images using Pyrender (Matthew Matl et al., 2020). For ShapeNet55 objects, we randomly create the train and test set using an 80-20 split; for chair and airplane classes, we cap the number of instances to 1,000. Since these objects have textures, we render them as RGB images using Blender (Community, 2018). For both datasets, we remove known symmetries from the rotation labels (i.e. for bottle, stool in ModelNet40 and bottle, bowl in ShapeNet55 we do not include rotations around z-axis).

### A.2. Baselines

**I2I** I2I uses a ResNet18-style encoder. The striding is set to 1 in the final two layers, resulting in an 8-by-8 feature map. This feature map is processed with a $1 \times 1$ convolution to project to the necessary dimensionality for the dynamic filter ($128 \times 7$). Convolutional layers

are instantiated with $C_4$-equivariant kernels. Since equivariant convolution layers include more trainable parameters than traditional convolution layers, we reduce the base width of the model from 64 to 38, such that it has similar model capacity to a traditional ResNet18 encoder (see Table 5). The equivariant convolutional layers use the regular representation, and a group pooling operation generates the trivial representation (e.g. traditional tensor) that is used during the projection. The projection to $V_{12}$ is performed with Gaussian kernels ($\sigma = 0.2$) such that the icosahedron's diameter covers 90% of the feature map. For training on object orientation prediction using Eqn. 4, we set $\lambda$ to 100. For training on shape classification, we perform average group-pooling on the output of the icosahedral group convolution to generate a probability distribution over the classes.

**CNN+GS** A ResNet-18 backbone with base width of 64 is used to produce a feature vector. This feature vector is processed with a linear layer to produce a 6D vector, which is converted to a valid $3 \times 3$ rotation matrix using the Gram-Schmidt process (see (Zhou et al., 2019) for details). The model is optimized using an L2 loss between the predicted and ground truth rotation matrices.

**CNN+Proc.** This uses the same architecture as CNN+GS, but outputs a 9D vector. This vector is converted to a valid rotation matrix using the Procrustes method (see (Brégier, 2021) for details). The model is trained using an L2 loss.

**CNN+IER** This method adds two additional linear layers to CNN+GS, which output rotational offsets. In other words, the first linear layer generates a rotation prediction, and the subsequent layers refine this prediction by additive residuals. The final output is converted to a valid rotation matrix using the Gram-Schmidt process and is optimized using L2 loss.

**CNN+$S_{exp}^3$** This method uses a ResNet-50 architecture whose linear layer outputs both a unit quaternion prediction and a class label. Vector normalization is used to produce a valid unit quaternion, and it is optimized using a cosine proximity loss. The class label is trained using a cross-entropy loss to predict which of the 8 quadrants of 3D space the quaternion lies in (3D because it only considers the imaginary components). For more details, see (Liao et al., 2019).

**E2CNN-Inv** The baselines E2CNN-Inv uses the same $C_4$ equivariant ResNet-18 architecture as I2I, but performs group-pooling to generate a $C_4$-invariant feature vector. This vector is processed with a linear layer to produce a probability distribution over classes.

**E2CNN-Eq** The baselines E2CNN-Eq uses the same $C_4$ equivariant ResNet-18 architecture as I2I. After the encoder, the representation is processed separately by: equivariant layers that predict rotations in the image plane; and invariant layers that predict arbitrary rotations. The output of the method is the product of the in-plane and out-of-plane rotation matrices. This idea of separating the prediction into equivariant and invariant components was initially suggested in (Falorsi et al., 2018) where the equivariance was enforced with regularization.

| Method | Num. Trainable Params |
|---|---|
| CNN (Gram-Schmidt) | 11.2 M |
| CNN (Procrustes) | 11.2 M |
| CNN $S^3_{exp}$ | 27.6 M |
| CNN+IER | 11.4 M |
| I2I (ours) | 10.9 M |

Table 5: Comparison of model capacity of baselines. The channels of I2I are reduced to compensate for the additional parameters of $C_4$-equivariant convolutional layers, resulting in a comparable model capacity of the non-equivariant architectures.

## Appendix B. Additional Results

### B.1. Comparing Projection Schemes

There are several possible approaches to mapping features from image space to the rotation manifold. I2I projects onto a submesh of the icosahedron to generate a dynamic filter (e.g. matrix-valued signal). We evaluate the pose prediction performance of different variations of I2I in Table 6. *Sparse projection* projects features onto the 6 vertices of the icosahedron (not the submesh); *full group projection* projects features onto the identity element of the $I_{60}$ group, rather than onto the homogenous space; *vector projection* projects features as vectors, and learns a matrix-valued feature sphere instead; I2Sphere is similar to I2I but projects onto the 2-sphere and performs spherical convolution. I2Sphere generates a signal over the continuous group, which is sampled using equally spaced grid and trained with a cross entropy loss.

| | Average Median Error (°) | |
| | 60 views | 15 views |
|---|---|---|
| I2I | 13.5 | 16.7 |
| I2I (*sparse projection*) | 12.7 | 17.8 |
| I2I (*full group projection*) | 19.2 | 56.1 |
| I2I (*vector projection*) | 14.1 | 22.1 |
| I2Sphere | 16.5 | 27.4 |

Table 6: Median rotation error (degrees) for different projection schemes, averaged over 10 object classes from ModelNet40. We include performance on fewer images since equivariance is especially important for sample efficiency. We find that the proposed method works well, while it is less important whether features are projected onto the icosahedron or a denser submesh of the icosahedron.

There are several interesting insights gained from the results in Table 6. First, projecting onto the icosahedron is better than projecting onto the full group. We hypothesize this is because the projection onto the sphere preserves spatial information about the location of features in the image. Next, we notice that using the discrete icosahedral group outperforms

the continuous group; however, additional effort should be put into refining the sphere approach since there are benefits to producing a continuous output.

## B.2. Effects of Object Centering

We test whether our method is shift-invariant, a necessary property in order to integrate it into multi-object 6D pose prediction pipelines. In such a pipeline, such as PoseCNN, the first step is to extract bounding boxes of relevant objects in the scene, which may have some positional noise. Thus, we test how robust I2I's orientation predictions are to object centering in the image by inserting random shifts at test time. In Table 7, we show results when I2I is trained on depth images of airplanes with different amounts shifting in and out of the camera plane. In-plane shifts are implemented by shifting the images by random numbers of pixels; out-of-plane shifts are produced by modifying the depth values of a pixels in an image. Shifts are performed on images in the training set and testing set. While highest accuracy is achieved with a centered object, we find that I2I's accuracy remains high even up to pixel shifts up to 30 pixels (for input images that are 128-by-128 pixels). We hypothesis the robustness is due to the shift-invariance of the ResNet projector, which has a downsampling factor of 16. These results suggest that our method could be trained end-to-end in conjunction with a bounding box prediction method such as Mask R-CNN.

| Pixel Shift | Median Error (°) | Depth Shift | Median Error (°) |
|:-----------:|:----------------:|:-----------:|:----------------:|
| ±5px  | 1.74 | ±5%  | 1.68 |
| ±10px | 1.76 | ±10% | 1.69 |
| ±15px | 1.90 | ±15% | 1.73 |
| ±20px | 2.02 | ±20% | 1.77 |
| ±25px | 2.14 | ±25% | 1.77 |
| ±30px | 2.32 | ±30% | 1.81 |

Table 7: Robustness of I2I to object translations evaluated on depth images of ModelNet40 airplane class. Results show median rotation error after 40 epochs. Both train and test sets are augmented with different amounts of pixel shifts (left) and depth shifts (right).

## B.3. Visualizing Orientation Predictions

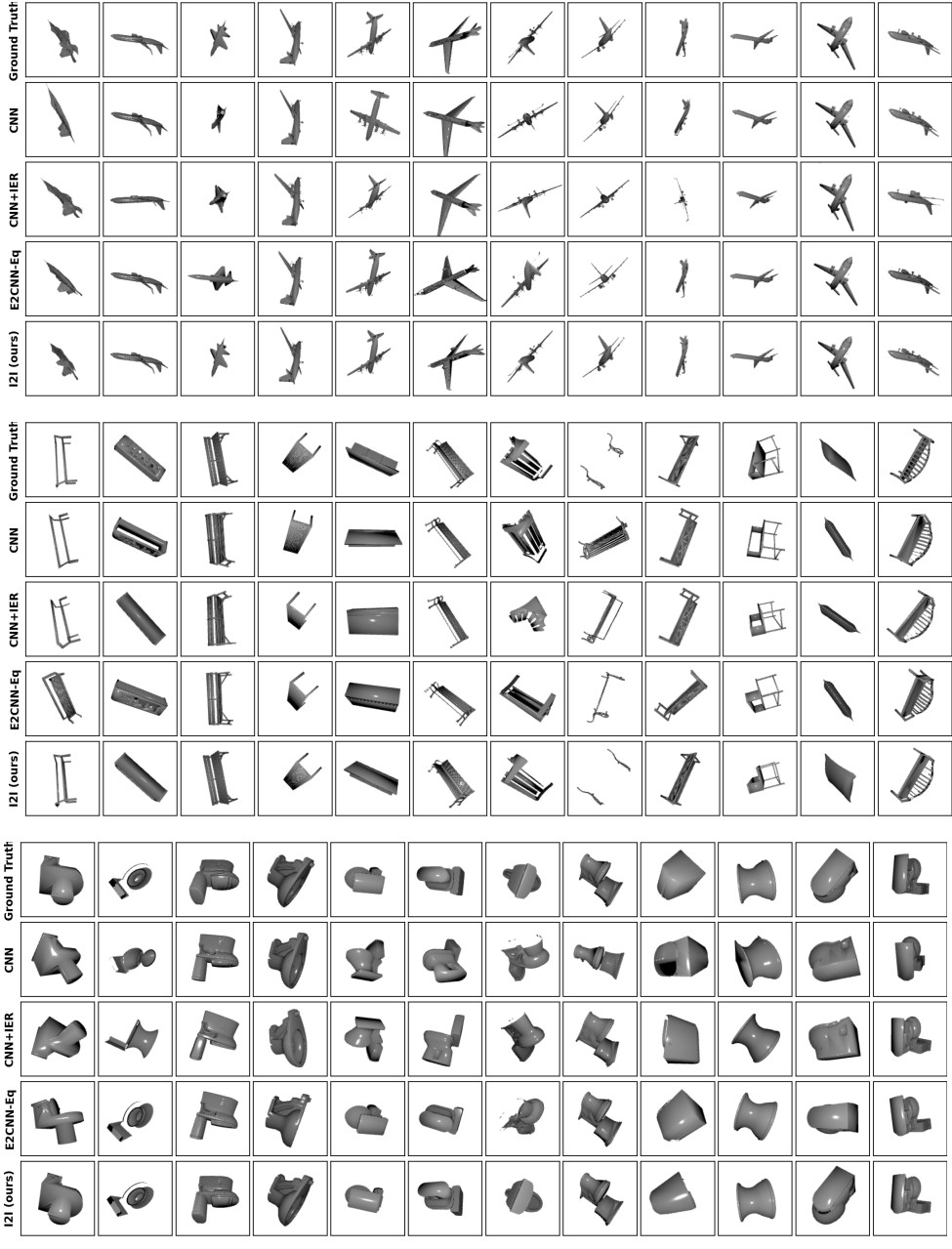

Figure 3: Example predictions for object orientation prediction task from Table **??** (grayscale images). To illustrate the predictions, we render new images of the respective objects using the SO(3) rotation predicted by each method. While I2I generally produces accurate predictions, large errors can be seen in columns 2 and 5 of the bench images.

## B.4. Representing Pose of Symmetric Objects

For this work, we did not consider cases where the object orientation is ambiguous due to symmetry or occlusion. While many works on object pose prediction rely on knowing symmetries in advance (to allow for symmetry-aware loss functions or manually disambiguating symmetries), we acknowledge that this assumption limits the applicability of our work. Even though the classification-regression framework of I2I cannot be used to model arbitrary or continuous object symmetries, we note that it is capable of modelling simple symmetries like the 180-degree symmetry of a dining table. To understand how this would work, refer to Equation 4: the regression loss is only applied to a group element that is correctly classified. Thus, the classification loss can capture uncertainty over object symmetries. Once trained, the model can represent a distribution over object symmetries by sampling predictions weighted by the probability assigned to each group element. Note, that this will fail for objects with continuous symmetries, e.g. water bottle, since there is not a single regression target associated with each group element.

