# OpenReview forum: "Image to Icosahedral Projection for $\mathrm{SO}(3)$ Object Reasoning from Single-View Images"
_NeurIPS.cc/2022/Workshop/NeurReps — NeurReps 2022 Poster_

### Official Review · Reviewer_kjak · 2022-10-13
**Interesting and sound method, with potential for improvement in the presentation and scope**

**Confidence:** 5
**Soundness:** 3
**Presentation:** 2
**Contribution:** 3
**Overall Rating:** 5

**Summary:**

The submission presents Image2Icosahedral, which is a method for predicting the class labels and orientation of 3d objects from single 2d images. It is based on three submodules: First, the image is encoded into a m*n channel feature map via a fully convolutional and planar rotation equivariant CNN. Secondly, these feature maps are then orthographically projected on (a mesh on) the non-occluded part of an icosahedron. Each feature vector is then reshaped into a m×n matrix, allowing to interpret the projected features as a dynamically predicted convolution kernel on the icosahedron. Lastly, this kernel is group-convolved with a learned and image independent feature map on the icosahedron. Rotation invariant features for classification are extracted by pooling over the elements of the symmetry group. For object orientation estimation, the authors propose to first classify the closest vertex, from which an offset rotation is subsequently regressed. This classification-then-regression approach makes the method more stable against large prediction errors, which can for instance occur when the object is symmetric and the orientation therefore only defined up to a coset.
The method is compared to several baselines. It performs better in an orientation prediction task than all other image based models and comparable to methods which are directly working on 3d point clouds. Its classification accuracy is worse than that of multi-view or point cloud based methods, however, these also have a richer input.

**Questions:**

see above

**Limitations:**

The authors address the limitation that the ground truth orientation is given as a point estimate, which is ambiguous if the model is symmetric.


**Recommended Decision:**

3: Accept

**Relevance:**

3: Solid fit

**Strengths And Weaknesses:**

The method is quite similar to Esteves' "Equivariant Multi-View Networks". The main novelties in comparison to this work are that the Image2Icosahedral needs only a single instead of |G| image embeddings ("multi-views") and that the image embedding is interpreted as a dynamic filter instead of a feature map on the icosahedron. The classification-then-regression approach is also novel and seems to improve the results considerably.

My main criticism is not in the method itself, which is technically sound, but in its presentation:
1) the authors claim that the method would SO(3) equivariant, which is not even possible for the considered task. Only the icosahedral correlation is SO(3) equivariant (or rather equivariant wrt a discrete subgroup). Due to the image rendering, there is no invertible SO(3) action on images. This is not a flaw of the method, but of the authors claims about its theoretical properties.
2) the paper keeps switching between referring to the continuous or discrete setting, for instance between SO(2) and C4, between the sphere and the icosahedron, or between SO(3) and its subgroup I60.
3) it is claimed that the method projects to a submesh on the icosahedron, but the filter is formally written as a map \psi: V_12 -> R^k on the non-subdivided icosahedron, which is confusing.
4) the submesh has 42 vertices, but it is not clear how these are chosen. They correspond most likely to the 12 original ico vertices plus subdivisions on the 30 edges, but this should be stated clearly.
5) equation 2 is actually not a group convolution, but what is sometimes referred to as a lifting convolutions (after which group convolutions follow). It is furthermore a correlation instead of a convolution.
6) more infos about the image embedding network should be given, for instance the e2cnn field types used and produced in the network output - are these scalar fields?

It would be interesting to consider a C5 equivariant image embedding with regular output feature maps. Projecting these on the icosahedron V12=I60/C5 would immediately yield regular group convolution features on the icosahedron. It would be nice to see both model variants in comparison.
One could furthermore include reflection equivariance in both the image embedding and icosahedral convolution.

**Submission Track:**

Proceedings Paper (9 Page)

---

### Official Review · Reviewer_GtPg · 2022-10-14
**A relevant topic and well executed**

**Confidence:** 4
**Soundness:** 3
**Presentation:** 3
**Contribution:** 3
**Overall Rating:** 7

**Summary:**

The paper proposes a new approach to learning 3D equivariant representations of 2D inputs. The proposed approach, Image2Icosahedral (I2I), learns a projection from the input image onto an icosahedron. By reasoning on this projection, the model is approximatively 3D equivariant. The paper assesses the model's performance by comparing it to various baselines, some reasoning on 2D inputs and some on partial point clouds. They find that I2I outperforms 2D input baselines on shape classification and inferring object orientation. They also found that it can outperform or equal point cloud methods with much less original information.

**Questions:**

Questions:

- Do you know why the I2Sphere might perform worse than I2I? Do you recognize that it might be due to less effort in optimizing it, or do you suspect a general problem?

- It would be interesting to compute training curves showing data efficiency of the different approaches (plot of the accuracy as a function of the number of training data). In particular, it would further demonstrate that reasoning in the discrete group can provide better performance.

**Limitations:**

The limitations are well adressed.

**Recommended Decision:**

3: Accept

**Relevance:**

3: Solid fit

**Strengths And Weaknesses:**

Strengths:

- The topic is relevant for computer vision, as reasoning in 3D with 2D inputs can be crucial in robotics.
- The paper is written, and the literature is well-referenced.
- A variety of baselines are included and well explained.
- Convincing approach and numerical results.

Weaknesses:

- While partially present in the appendix, a better motivation to use the quotient group $V_{12}$ instead of the whole group would be beneficial.

**Submission Track:**

Proceedings Paper (9 Page)

---

### Official Review · Reviewer_iNtK · 2022-10-16
**Image to Icosahedral Projection for SO(3) Object Reasoning from Single-View Images**

**Confidence:** 4
**Soundness:** 3
**Presentation:** 3
**Contribution:** 2
**Overall Rating:** 6

**Summary:**

Due to differences in appearance brought on by seeing the object from various orientations, it can be difficult to infer information about 3D objects from 2D photographs. Other tasks, such pose estimation, are equally invariant to 3D rotations as tasks like object classification. However, because we lack an a priori model of how the image changes in response to out-of-plane object rotations, enforcing equivariance as a model constraint is frequently not feasible with 2D image input. The only existing SO(3)-equivariant models require point cloud or voxel input instead of 2D pictures .In this study, authors propose a unique architecture that reasons in SO(3) by learning a projection of the input image onto an icosahedron and is based on icosahedral group convolutions. The model that results is about equivariant to rotation in SO (3). They test their model on challenges including object position estimation and shape classification and discover that it performs better than expected baselines.

**Questions:**

Describe more in details who you would mitigate the fact that you cannot describe ambiguities in orientation for objects with symmetry.

**Limitations:**

Model is limited by the task to relying on learned symmetry in the icosahedral filter projector as the group action is unknown in image space.
Cannot describe ambiguities in orientation for objects with symmetry.

**Recommended Decision:**

2: Borderline

**Relevance:**

3: Solid fit

**Strengths And Weaknesses:**

Strength:
A novel method, Image2Icosahedral, for learning 3D representations of objects from single-view 2D images, coupled with SO(3)-equivariant techniques.
Test on existing datasets.

Weaknesses:
Model is limited by the task to relying on learned symmetry in the icosahedral filter projector as the group action is unknown in image space.
Cannot describe ambiguities in orientation for objects with symmetry.

**Submission Track:**

Proceedings Paper (9 Page)

---

### Decision · Program_Chairs · 2022-10-21

Accept (Poster)